# Understanding the health challenges of Amazonian riverine communities: A qualitative study on community perceptions amid climatic change

Alícia Patrine Cacau dos Santos[1,2], Theo de Lima Guerra[1,3],
Evellyn Antonieta Rondon Tomé da Silva[1,2], Inácio Coelho Sá[1,3],
Hiran Sátiro Souza da Gama[1,2], Ana Paula Silva Oliveira[1], Rafaela Nunes Dávila[1,2],
Hélio Afonso Amazonas Júnior[1], Joaquim da Silva Lopes[4], Esly Camico Mandu[3],
Tatiana Bacry Cardoza[5], Jaqueline de Almeida Gonçalves Sachett[1,2],
Fernando Almeida-Val[1,2,3], Altair Seabra de Farias[1,2], Vinícius Azevedo Machado[1,2],
Wuelton Marcelo Monteiro[1,2] Felipe Leão Gomes Murta[1,2]*

**1** Fundação de Medicina Tropical Doutor Heitor Vieira Dourado, Manaus, Amazonas, Brazil,
**2** Universidade do Estado do Amazonas, Manaus, Amazonas, Brazil, **3** Universidade Federal do
Amazonas, Manaus, Amazonas, Brazil, **4** Instituto Nacional de Pesquisas da Amazônia, Manaus,
Amazonas, Brazil, **5** Centro Universitário Fametro, Manaus, Amazonas, Brazil

* felipelmurta@gmail.com

journal.pone.0333408

Mississippi, BRAZIL

**Peer Review History:** PLOS recognizes the
benefits of transparency in the peer review
process; therefore, we enable the publication
of all of the content of peer review and
author responses alongside final, published
articles. The editorial history of this article is
available here: https://doi.org/10.1371/journal.
pone.0333408

## Abstract

### Background

In the Brazilian Amazon, approximately 900,000 people live along riverbanks, rely-
ing on the biome for survival and livelihoods. Severe droughts and flooding events,
intensified by the El Niño phenomenon and human-driven actions such as defor-
estation and fires, exacerbate the isolation of traditional communities. This isolation
limits access to essential healthcare services and worsens existing health inequalities
among vulnerable populations. This study investigates how extreme climatic events
impact health and healthcare access in traditional riverine communities, focusing on
their perceptions and lived experiences.

### Methodology/principal findings

A qualitative study was conducted in a riverine community in Tabatinga, Western
Brazilian Amazon. Thirty-two purposively selected participants (farmers, fishermen,
and students) took part in in-depth interviews and focus-group discussions. The-
matic analysis was conducted using MAXQDA software. After the analysis two main
themes emerged: (1) the impacts of Amazonian seasonality on healthcare access,
and (2) the dynamics of arrivals and departures along riverbanks. Participants high-
lighted health challenges tied to seasonal extremes, including water and food inse-
curity, financial vulnerability, spikes in gastrointestinal diseases, increased accidents

**Data availability statement:** All relevant data are within the manuscript and its Supporting Information files.

**Funding:** Financial support was provided by grants from the Fundação de Amparo à Pesquisa do Estado do Amazonas (FAPEAM, Manaus, AM, Brazil: Grant no. 018/2023 - PROCLIMA-AMAZÔNIA, awarded to F.A-V.) and POSGRAD UEA FAPEAM (Grant no. 008/2021, awarded to A.P.C.S.). The funders had no role in study design, data collection, analysis, decision to publish, or manuscript preparation.

**Competing interests:** The authors have declared that no competing interests exist.

with aquatic or wild animals, river pollution, and higher drowning rates. They also noted births and deaths occurring along riverbanks during extreme events. Most respondents perceived that extreme weather events, especially severe droughts, have intensified over the past decade, worsening health and food/water security and further restricting healthcare access.

## Conclusions/significance

In this scenario, riverine communities are highly vulnerable to climate change, as existing health challenges are compounded by escalating climatic crises. Tailored healthcare solutions, including telemedicine platforms, mobile clinics, and resilient transportation networks, are urgently needed. Investments in communication infrastructure and emergency air transportation are critical as riverine navigation becomes increasingly unreliable.

## Introduction

The Amazon Rainforest is recognized as the largest remaining tropical forest on Earth. However, this biome faces serious threats from deforestation, illegal fires, and the intensifying extreme climatic events [1,2]. The forest spans nine countries in South America; Brazil contains 60% of its total area, equivalent to 49% of Brazil's territory [3]. In these regions, approximately 951,867.24 riverine people live along the banks of Amazonian rivers and their tributaries [4]. These communities built their livelihoods on these rivers, developing a deep connection to the natural environment through the use of the forest and river resources [5].

Amazonian rivers serve as the main routes for transportation and subsistence and help shape the cultural identity of local populations [5,6]. In the state of Amazonas, the Solimões River is especially important. Originating in Peru, it crosses into Brazil at Tabatinga and flows approximately 1,600 km before merging with the Rio Negro to form the Amazon River, which empties into the Atlantic Ocean [7]. Along its course, the Solimões River is divided into Alto (Upper), Médio (Middle), and Baixo (Lower) Solimões regions. The Upper Solimões region serves as an economic hub, encompassing Tabatinga (Brazil), Leticia (Colombia), and Santa Rosa (Peru), a district in the province of Mariscal Ramón Castilla. Tabatinga and Leticia form a transboundary hub characterized by commercial, cultural, and social exchanges [8].

In recent years, Amazonas has experienced severe droughts driven by rising temperatures, deforestation, fires, and an intensifying El Niño phenomenon, further aggravated by human activities [2,9–11]. In this context, Indigenous peoples, and Quilombolas are most affected, routinely contending with geographic isolation, food insecurity, and water scarcity [12]. The droughts have severely disrupted river transportation, a critical means of accessing food, goods, healthcare and other essential services [13]. Moreover, this situation has heightened the geographic isolation of certain communities, significantly extending response times for health emergencies and exposing lives to greater risks, particularly for those requiring continuous care

[11]. A study highlighted the challenges riverine populations face in accessing healthcare: they travel an average of 60.4 km and take approximately 4.2 hours to reach urban centers with available services [14]. These findings illustrate how the distance and difficulties faced by these communities could be further intensified by the worsening global climate crisis [15]. Although existing research has effectively quantified transportation barriers and travel times, significant gaps remain in understanding how local residents perceive and respond to the climate-mediated health risks associated with increasingly frequent droughts and floods. The growing body of literature on health-seeking behaviors among vulnerable populations, including studies employing the Health Belief Model (HBM) to examine risk perception and care accessibility, demonstrates how environmental stressors amplify health disparities [16]. Within Amazonian contexts specifically, studies have revealed the role of cultural frameworks in shaping risk interpretation and response strategies in healthcare [17–19]. Thus, this study aimed to document how seasonal droughts and floods affect healthcare-seeking behaviors and understand residents' perceptions of climate change over the last years. This provides unique qualitative insights and fills a critical gap in understanding health disparities in remote Amazonian communities.

## Materials and methods

### Ethical statement

Ethical approval for this study was obtained from the Institutional Review Board of the Fundação de Medicina Tropical Dr. Heitor Vieira Dourado (CAAE: 40850020.1.0000.0005), ensuring compliance with established ethical guidelines for research involving human participants. The project was originally approved focused on educational interventions for snakebite prevention, however one of its core objectives was to understand the main health challenges faced by the community. Within this framework, the climate change theme emerged organically during fieldwork, proving intrinsically connected to participants' health reports an alignment that was already contemplated in the study objectives. To ensure ethical compliance, we implemented: additional oral clarifications whenever new climate related aspects emerged; and oral revalidation of informed consent for data use, maintaining ethical principles throughout the investigative process. Data collection and recruitment occurred between 1st June 2022 and 20th December 2022. Written informed consent was obtained from all adult participants. Written informed consent was obtained from adult participants, and parental or guardian consent was obtained for participants under 18. All adult participants were informed about the study's objectives, risks, and benefits using accessible language. Capacity to consent was presumed for participants over 18 years old without known cognitive impairments, while minors provided verbal assent with written consent from parents/legal guardians. Confidentiality was rigorously maintained by anonymizing data during transcription and securely storing all materials. Local norms and cultural sensitivities in the tri-border area were carefully observed and respected to foster trust while safeguarding participants' dignity and autonomy. The study adhered to the COREQ guidelines (Consolidated Criteria for Reporting Qualitative Research) to ensure high-quality reporting of the research process and findings. See (S1 file) for COREQ items from Domain 1–3 [20].

### Inclusivity in global research

Additional information regarding the ethical, cultural, and scientific considerations specific to inclusivity in global research is included in the supporting information. See (S2 File).

### Study area and population

The study was conducted in a riverine community in Tabatinga, a city in the far west of Amazonas State, Brazil, within the Alto Solimões region (Fig 1).

Over the past few years, the Alto Solimões region has faced severe droughts threatening its rich biodiversity and the cultures of diverse peoples. These communities face significant challenges in accessing basic services essential for

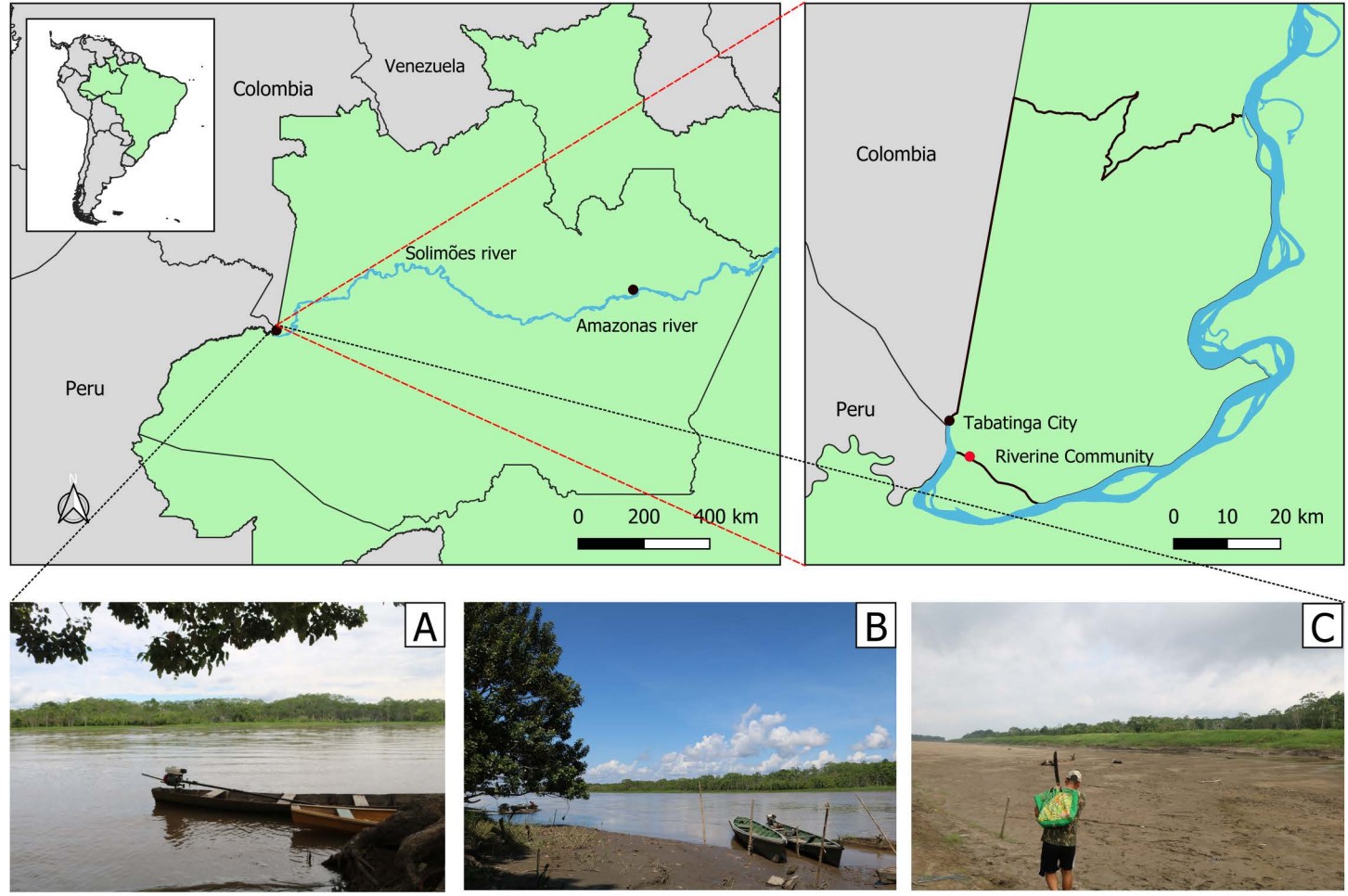

**Fig 1. Area of study.** A: Tributary of the Solimões River during the Amazonian winter; B: Tributary of the Solimões River at the beginning of the dry season; C: Tributary of the Solimões River during the Amazonian dry season. The base used to create the map is from the IBGE (Brazilian Institute of Geography and Statistics), which is freely accessible for creative uses in shapefile format, following the Brazilian Access to Information Law (12,527/2011) [21].

well-being, such as healthcare, perpetuating social inequalities [22,23]. The tri-border location where Brazil, Colombia, and Peru meet increases challenges in accessing healthcare services. These issues arise from population movement across borders, communication difficulties between the countries' healthcare systems, and criminal activity. This situation reinforces cycles of poverty and social inequality, with an infant mortality rate higher than the Brazilian national average.

The riverine community in this study, like many in the Brazilian Amazon, experiences marked seasonality, alternating between two distinct periods known as the Amazonian winter and summer. The Amazonian winter begins in mid-October and lasts for approximately eight months, characterized by torrential rains and rising river levels. During this period, boats serve as the primary mode of transportation used for agricultural and fishing activities, education, healthcare, and public safety. In contrast, during the Amazonian summer, as river levels recede, boats are largely replaced by motorcycles, off-road vehicles, and walking. This period intensifies geographic isolation, affecting numerous riverine communities along the Solimões River.

The study included individuals aged 12 years or older who were permanent residents of the riverine community and engaged in subsistence activities (small-scale farming, fishing, or hunting) that are directly affected by extreme climate

events and voluntarily agreed to participate were eligible. The minimum age of 12 years was selected based on two criteria: (1) this marks the developmental stage when children begin participating meaningfully in family subsistence activities, and (2) younger children typically do not engage in these livelihood practices to the same extent. No participants withdrew consent during the study. Individuals under 12 were not included because the interviews and group discussions required communicative abilities typically found in participants aged 12 years and older. Participants were recruited through purposive sampling, and the principle of theoretical saturation [24] guided data collection, ensuring clear patterns emerged across interviews and discussions.

## Data collection

In-depth individual interviews (IDIs) and focus group discussions (FGDs) were held between June and December 2022 using a semi-structured interview guide. This guide, developed by the research team, included open-ended questions, allowing the interviewer to explore topics in greater depth as needed (Table 1). The guide was previously validated with a smaller sample of volunteers sharing the same characteristics as the study population to refine the language and ensure clear understanding of the questions.

The IDIs and FGDs were conducted face to face by the qualitative researchers [APCS (Alícia Patrine Cacau Dos Santos, PhD), EARTS (Evellyn Antonieta Rondon Tomé Silva, master's degree student), and HSSG (Hiran Sátiro Souza da Gama, master's degree student)],all healthcare professionals and field researchers with training and experience in qualitative data collection. The IDIs and FGDs were scheduled according to participant availability, with an average duration of 40 minutes for IDIs and 60 minutes for FGDs. Both were conducted in quiet, comfortable locations such as participants' homes, schools, or church community buildings, where only researchers, observers, and participants were present. Data generated through the IDIs and FGDs were audio-recorded, documented in field notebook entries, and supplemented by participant observations conducted by the team, enabling data triangulation during analysis.

## Data analysis

The data were transcribed without personal identifiers and uploaded into MAXQDA software (version 2020) for reflexive thematic analysis, following Braun and Clarke's approach [25]. Inductive coding was conducted line-by-line by two researchers and reviewed (S3 File). Then preliminary categories were developed and refined through consensus, culminating in the final themes. The analysis triangulated data from in-depth interviews (IDIs), focus group discussions (FGDs), field notebook entries, and participant observations to provide a detailed understanding of healthcare access challenges in the region. Methodological rigor was ensured through reflexive thematic analysis and data triangulation. Participant diversity, including men, women, and adolescents, enriched the findings by capturing a wide range of community experiences.

**Table 1. Participant's interview guide.**

| Questions |
| --- |
| 1. What are the main health issues in the community? (considering the dry season, the flood season, and the transition period as the water levels begin to rise) |
| 2. What are the main challenges regarding access to healthcare? |
| 3. What do people in the community do when facing health problems? |
| 4. Describe how you would reach a healthcare facility. |
| 5. Which healthcare facility would you seek out? |
| 6. Is there anything that would prevent you from seeking medical help? Please describe yourself. |
| 7. If a serious health issue arises in the community, what do you do? (e.g., heart attack, childbirth. What is the process?/ How do you communicate?/ Do you spend money?) |

### Research team and reflexivity

The study team comprised seven researchers with doctoral degrees and expertise in qualitative research and tropical and infectious diseases, including two women (APCS and JAGS) and five men (FLGM, VAM, WMM, FAV and ASF). Additionally, the team included seven PhD-qualified qualitative researchers, five master's degree students, and five fieldwork researchers. To minimize the influence of their perspectives on data collection and analysis, the team made a conscious effort to remain objective and neutral. The study team had no prior interactions with the study participants.

## Results

The study included a total of 32 participants (Table 2), comprising 15 in-depth individual interviews and 3 focus group discussion participants across three distinct groups: a women's group (n = 5), a men's group (n = 5), and an adolescent group (n = 7). Participants. During early data collection, researchers observed that gender and age dynamics could influence participant responses. To account for this, focus groups were conducted separately.

The data analysis identified two main themes. The first, Impacts of Amazonian Seasonality on Healthcare Access for Riverine Communities, explores the challenges in seeking medical assistance during both dry and flood seasons. The second, Arrivals and Departures Along the Banks of the Solimões River, addresses women's health at various life stages—pregnancy, childbirth, and postpartum and includes accounts of stillbirths during these journeys.

### Theme 1: Impacts of Amazonian seasonality on healthcare access for riverine communities

Participants emphasized how seasonal extremes affect health in the community. They reported barriers to healthcare access, water and food insecurity, accidents with aquatic animals, more wildlife encounters, financial vulnerability, increased gastrointestinal illnesses, worsening river pollution, and drowning incidents, distributed in two sub-themes, which together demonstrates how seasonality directly shapes healthcare-seeking decisions.

**Sub-theme 1a: Impact *of the Amazonian dry season on health*.** During the dry season, geographic isolation worsens, creating major barriers to healthcare access. Community members often travel long distances under intense heat to reach riverbeds, and boat transport is further hindered by the need to manually navigate through shallow waters. These challenges particularly impact those needing specialized care. Participants noted that emergency and urgent care teams struggle to reach their communities.

*"When this area in front of the community is in the dry season, the entire river turns into a beach. To reach the riverbed, we first have to cross the beach that forms, then a dense patch of vegetation, and finally another large beach. Depending on a boat during this time feels like we're risking death"* (Participant 10, female, general services, 39 years old).

**Table 2. Participant's characteristics.**

| Occupation | Men (n = 16) | Women (n = 16) | Total (n = 32) |
|---|---|---|---|
| Farmer | 4 | 7 | 11 |
| Fisherman | 5 | 0 | 5 |
| Stay-at-home spouse | 0 | 4 | 4 |
| Housekeeper | 0 | 2 | 2 |
| Teacher | 2 | 1 | 3 |
| Student | 4 | 2 | 6 |
| Community health worker | 1 | 0 | 1 |

Boat-based mobile clinics must pause until levels rise again. Consequently, individuals with chronic conditions often stay longer in urban centers, causing financial and emotional strain from being far from their families. Beyond fuel costs, many riverine residents also pay for in-city transportation, lodging, food, medications, and other necessities.

*"We spend a lot of money trying to get healthcare, especially on gasoline and medication. Sometimes we don't have enough money, and the healthcare professionals tell you to go to the clinic, but when you get there, they don't have the medicine either"* (Participant 3, female, farmer).

Droughts exacerbate water insecurity because rain, a primary drinking water source, becomes scarce. Fires also contaminate rainwater with smoke, making it dark and smoky tasting, which leads to more gastrointestinal illnesses.

*"During the summer, it can go months without raining, and we rely on rainwater, so we end up with no water often [...] When it hasn't rained for a long time, the roof tiles get dirty, so when we collect that water to drink, it might carry some bacteria or viruses, causing diarrhea, nausea, vomiting, and even skin itching after we consume it. On top of that, it tastes like smoke because they burn [the forest] a lot in the summer, so when it rains, the water ends up dark"* (Participant 3, female, farmer).

*"When there's smoke here and it rains, the water tastes like smoke. I don't like that smoky-tasting water; the water turns black"* (Participant 25, female, student).

*"In the dry season, it sometimes goes 15 to 20 days without raining. So, when it does rain and we store the water, it always becomes very dirty; it's dark water, even though it's rainwater. We store it in large containers, and when we check, the water is black. Therefore, using this water for consumption and drinking is very challenging"* (Participant 8, female, farmer).

Participants reported that intense drought increases their contact with aquatic animals, particularly spiny fish, resulting in frequent accidents when moving boats through newly formed small water channels.

*"When the river dries up, fish sometimes get stuck in lagoons, and accidents involving their spines occur. I, for instance, have stepped on a spiny fish and been shocked by an electric eel [Electrophorus electricus]"* (Participant 5, male, fisherman).

These recurring isolation and transportation challenges during the dry season emphasize the need for tailored healthcare delivery. For instance, strategies such as satellite clinics equipped with telemedicine capabilities and the deployment of solar-powered water purification units could address the dual issues of healthcare access and water insecurity.

**Sub-theme 1b: Impact of the Amazonian rainy season on health.** The rainy season increases health and safety risks due to more encounters with displaced wildlife, such as jaguars, caimans, and anacondas. Seasonal floods also raise child drownings rates, exacerbated by home features like openings in wooden floors (used for daily activities) that pose fatal hazards. Together, these challenges show how the rainy season compounds risks, especially for children and other vulnerable groups.

*"During the floods, animals look for dry land. For example, in another community, they found a snake inside the bedroom where wet clothes had been left"* (Participant 13, male, teacher).

*"During the Amazonian rainy season, a child in a neighboring community drowned because the house is surrounded by the river, and there's nowhere else to go"* (Participant 5, male, fisherman).

Another major issue is food insecurity: climate variability disrupts seasonal weather patterns and directly affects subsistence farming. For instance, when water levels rise earlier than expected, entire harvests can be ruined.

*"The climate has changed. It's not the same anymore. I used to know when the river would rise and recede, but now it fills up and dries out at the wrong times. This affects us because we plant crops to harvest near the flood season, and if the water rises earlier than expected everything rots, that's a loss for us. On top of that, the fish aren't following the same patterns. It used to be good for fishing when the river started drying up because the fish would come out from everywhere, but now they don't"* (Participant 22, male, 19 years old).

*"The characteristics of seasonality have completely changed. Before, we used to have a garden and could estimate when to harvest, but now, with this climate, it's become very difficult to calculate. I used to grow peppers, green onions, and scallions in my garden, but now, with the extreme rain and drought, they die. The same happens with tomatoes. Because of this, this year I decided to plant them in pots inside the house"*(Participant 3, female, farmer).

*"When I was younger, I had a clearer understanding of the climate, but nowadays, everything has changed. For example, floods used to occur in December, but this year the season is still dry. This makes it impossible to plan plants properly. Corn, for instance, is a fast-growing crop and can be harvested in three months, while cassava and bananas take seven to eight months. However, it all depends on the water levels; if the water comes earlier or later, it directly affects the harvest timing"* (Participant 5, male, fisherman).

Riverine residents identified river pollution as another major challenge. Contaminants include dead animals, human waste, plastics, metal utensils, furniture, and heavy metals, much of which enters from Brazil, Peru, and Colombia. This pollution affects health through contaminated fish and disrupts subsistence fishing when debris entangles gear.

*"River pollution is terrible in our area, and I believe throughout the entire Solimões River, as it's a border region. Here in Tabatinga, we take extra care to avoid polluting the water, but I'm unsure about Peru and Colombia. Based on the trash that reaches us, it seems they don't pay much attention to protecting nature"* (Participant 8, female, farmer).

*"We catch fish from the river, but all that rotten stuff in the water, like food scraps and dead animals, affects the fish [...]. Then we end up catching sick fish"* (Participant 10, female, general services).

High child drowning rates during the rainy season highlight the need for targeted public health measures, child safety education, emergency flotation device distribution, and housing adaptations (e.g., secure barriers for open flooring). To address food insecurity, promote climate-resilient farming and establish community seed banks. Partnerships should also focus on infrastructure improvements: better waste management to reduce river pollution and flood-adapted housing designs. These measures would strengthen resilience and well-being of riverine communities during the rainy season.

## Theme 2: Arrivals and departures along the riverbanks

This theme reflects the combined impact of several factors discussed previously, including seasonality, geographic barriers and isolation, financial vulnerability, compounded by precarious healthcare infrastructure. These challenges often define the journey to health facilities, where "arrivals" mean births and "departures" represent deaths, both outcomes shaped by the difficulties in accessing medical care.

*"I once saw a woman give birth on the riverbank while we were heading to Tabatinga. I remember her husband spreading out a sheet on the sand, and she delivered the baby right there. Later, the medical team arrived and took the baby, all wrapped up"* (Participant 32, female, farmer).

*"My cousin from another community was pregnant with twins, and since there wasn't enough time to reach the hospital, the baby who was supposed to be born first died during the journey"* (Participant 8, female, farmer).

In this context, women across age groups face geographical isolation, that makes medical care difficult to access. Routine exams, such as cervical cytology, vaginal ultrasounds, and mammograms, become unreachable. Moreover, they report being unable to use preferred contraceptive methods due to difficulty in reaching the urban center of Tabatinga.

Women's health is further impacted by limited prenatal care from a healthcare system that does not adapt to the realities of riverine communities. This often forces women to migrate to urban areas for better care. However, such migration creates vulnerability, as it requires financial resources that may not be available. Consequently, women without the necessary means are left stranded along the dry or flooded riverbanks, walking or navigating, and relying on experienced peers for childbirth assistance.

*"Women from nearby riverine communities came here because they couldn't make it to the hospital, and the delivery was handled by the community midwife"* (Participant 29, female, farmer).

*"We always encourage pregnant women to go to the city when their due month arrives; they have to stay at a relative's house"* (Participant 9, male, farmer).

These challenges highlight critical gaps in maternal healthcare infrastructure. A mobile maternal health program offering routine prenatal visits and delivery services could mitigate risks of emergency births along riverbanks. Such programs should also integrate culturally sensitive care to address the unique needs of riverine populations.

## Discussion

This study highlights how the unique climatic conditions of the Amazon significantly influence the health and well-being of riverine populations. Seasonal extremes, including geographic isolation, food and water insecurity, increased health risks, and healthcare access barriers, pose compounding challenges for these communities. In the Peruvian Amazon for example, flood seasons replicate these patterns, particularly through increased exposure to venomous animals, heightened risk of child drownings, as reported by our participants, and severe impacts on subsistence farming in Peru [26]. During extreme droughts, regional evidence confirms systematic links to food insecurity, water source contamination, and potable water shortages, creating critical sanitation crises [27]. These findings reveal the critical interplay between environmental factors and healthcare access, showing how climate variability exacerbates vulnerabilities. For instance, the contamination of water sources during droughts and floods leads to increased gastrointestinal diseases, while geographic isolation during extreme events severely limits access to emergency care. Addressing these compounded vulnerabilities underscores the essential role of environmental health in improving resilience and mitigating disparities in climate-impacted populations.

While our qualitative data focus on Tabatinga, these experiences mirror those of vulnerable riverine communities worldwide. Indeed, the World Health Organization (WHO) estimates that 3.6 billion people live in regions highly vulnerable to extreme weather events and climate variability. In these areas, it is projected that between 2030 and 2050, climate-related impacts will lead to approximately 250,000 additional deaths annually, primarily due to malnutrition, malaria, diarrhea, and heat stress [28]. Regions with inadequate health service infrastructure are expected to be disproportionately affected [29].

The study revealed that seeking care largely dependent on riverine transportation has consistently faced challenges from seasonal fluctuations, financial constraints, and long distances [13]. These issues have been worsened by the climate crisis, making essential needs (potable water, food security, and basic medical services) increasingly out of reach [29,30]. These implications extend to vulnerable populations worldwide. Communities dependent on natural resources and river systems face similar challenges: waterborne diseases, disrupted livelihoods, and geographic isolation due to climate

change. Addressing these issues requires integrated solutions that balance environmental sustainability and public health needs.

As a result, millions may face food insecurity due to these communities' heightened economic vulnerability. Many rely on fishing and stable temperatures for subsistence farming. The lack of nutritious, safe food undermines health, leading to more infectious and chronic diseases, malnutrition, and weakened immunity, particularly among children and the elderly. This food insecurity also aggravates the prevalence and severity of neglected diseases in the region, perpetuating a cycle of vulnerability and deteriorating health conditions.

Droughts and floods also hinder potable water access, leading to more gastrointestinal diseases among children, potentially fatal and contributing to high or rising infant mortality rates [31,32]. Evidence suggests that global changes may accelerate infectious disease emergence by disrupting ecosystem balance [33].

Addressing future health challenges requires healthcare systems to adapt to barriers posed by extreme climatic events. Priority actions include deploying mobile healthcare units, enhancing transportation networks, and expanding telemedicine platforms tailored to remote riverine communities. These must address the rising burden of injuries, deaths, and infectious diseases linked to extreme weather and unsafe water and food. Tackling food and water insecurity is equally critical to reducing long-term health impacts on vulnerable populations [34].

According to the 2022 Climate Inequality Report [35], climate change risks and impacts are unevenly distributed, intensifying social inequalities, particularly in access to healthcare. This dynamic contributes to worsening Brazil's already low primary care coverage rates [17,19,35]. Addressing situations like those described here will likely become more common, as populations and healthcare systems struggle with accelerating changes.

Involuntary relocation, as observed with pregnant women in this study, is a recurring reality, especially in the pursuit of better access to healthcare and education. Projections suggest that displacement will increase in coming decades; however, this phenomenon is often tied to precarious living conditions, with higher risks of violence and insecurity [36,37]. For riverine communities, displacement also causes profound disruptions and may even erase their way of life, since identity is intrinsically tied to their connection with nature [38].

To prevent thousands of deaths, WHO [28] outlines three key objectives: reduce carbon emissions while improving health outcomes; build stronger, climate-resilient, environmentally sustainable healthcare systems; and safeguard health from climate change impacts. Consequently, healthcare systems must prioritize enhancing access, particularly addressing the unique needs of traditional riverine communities.

As of 2020, only 13 river ambulances served all urgent and emergency cases nationwide—far fewer than needed for the riverine population [39]. Additionally, river transport is increasingly impractical due to prolonged droughts, underscoring healthcare systems' urgent need to adapt. Climate change is escalating into the most significant threat to human health in the 21st century [40].

As a theoretical framework for addressing the health challenges we indicate the use of Ostrom's approach, especially for governmental organizations [41]. The framework provides a robust foundation for mapping interactions among socio-ecological variables. It enables the identification of vulnerability factors as well as opportunities for local public policies that integrate traditional knowledge and institutional innovation. By doing so, this approach avoids generic solutions, instead promoting adaptive interventions tailored to the actual needs of riverine populations in the face of climate change.

Besides adapting the healthcare system indicated by WHO, it is essential to consider the concept of ancestral futures, a perspective advocated by the Indigenous movement in Brazil and championed by Ailton Krenak [42], a prominent Indigenous thinker. This concept emphasizes the value of Indigenous knowledge in shaping strategies to address this critical moment. A paradigmatic example of this synergy can be observed in sustainable environmental management, where Indigenous knowledge plays a decisive role in ecosystem preservation. It is crucial to understand that the health of traditional communities is intrinsically tied to socio-environmental justice, revealing the interdependence between human well-being and ecological balance. This holistic vision requires integrating diverse knowledge systems. As demonstrated

by [43] on moringa seed filtration of contaminated water, a practice elders identified as historically effective, such techniques could be incorporated into mobile clinic protocols. This would both immediately reduce gastrointestinal illnesses and validate traditional knowledge systems. However, it is important to remember that the water is filtered because the river is sick, but the cure comes from the standing forest. Thus, while moringa filtration is a crucial emergency measure, the structural solution requires climate justice: protecting springs, combating illegal mining, restoring degraded areas, and addressing the root causes of water scarcity through land rights and fire prevention policies. Within this epistemic landscape, a recent study conducted by indigenous researchers highlighted similarities and differences between modern scientific knowledge and Indigenous Amazonian knowledge. It proposes alternatives for dialogue and integration, combining modern conservation science strategies with Indigenous knowledge, as a pathway to the regional sustainability [44].

Due to the specific cultural and geographical characteristics of the riverine community studied and considering that data were collected inly from June to December (a single weather cycle) the results may not reflect the experiences and perceptions of all Brazilian Amazon riverine communities. Additionally, nuances of local expressions may have been lost in translation from Portuguese to English.

## Conclusions

This study highlights the profound health impacts of climatic events on traditional riverine communities shaped by Amazonian seasonality. The increasing frequency and severity of droughts and floods disrupts daily life, exacerbating healthcare access, food security, and transportation challenges. Urgent measures are needed to implement adaptable healthcare systems and sustainable infrastructure, including telemedicine platforms, mobile clinics, and resilient transportation networks. Investments in communication infrastructure and air emergency transportation are critical as riverine navigation becomes increasingly unreliable. Because participants described waiting days for boats or relying on relatives for childbirth, these interventions directly address their most pressing barriers. Embracing the concept of ancestral futures emphasizes humanity's interconnectedness with the environment and provides a framework for sustainable solutions. Sustainable environmental and water resource management in the Amazon for example, when conducted with ecological balance and respect for biome limits, serves as a critical strategy for preventing the emergence and spread of socially determined diseases. By mitigating environmental degradation and natural resource depletion, key drivers of health vulnerabilities, this approach supports the maintenance of functional ecosystems capable of regulating disease vectors; the preservation of water sources meeting qualitative standards for human consumption; and the sustainability of traditional livelihoods intrinsically linked to collective health.

These strategies can enhance resilience, improve the quality of life for riverine populations, and preserve their cultural and environmental heritage while serving as a model for addressing similar challenges in other climate-vulnerable regions. Although our findings come from one tri-border community, future studies should examine other Amazonian regions to refine these recommendations.

## Supporting information

**S1 File. COREQ.** Consolidated criteria for reporting qualitative studies.
(DOCX)

**S2 File. Checklist.** Inclusivity in global research.
(DOCX)

**S3 File. Codebook from qualitative analysis.**
(PDF)

## Acknowledgments

We extend our gratitude to all riverine participants, whose contributions were essential to the development of this study.

## Author contributions

**Conceptualization:** Alícia Patrine Cacau dos Santos, Theo de Lima Guerra, Evellyn Antonieta Rondon Tomé da Silva, Inácio Coelho Sá, Hiran Sátiro Souza da Gama, Ana Paula Silva Oliveira, Rafaela Nunes Dávila, Hélio Afonso Amazonas Júnior, Tatiana Bacry Cardoza, Jaqueline de Almeida Gonçalves Sachett, Fernando Almeida-Val, Vinícius Azevedo Machado, Wuelton Marcelo Monteiro, Felipe Leão Gomes Murta.

**Data curation:** Alícia Patrine Cacau dos Santos, Theo de Lima Guerra, Evellyn Antonieta Rondon Tomé da Silva, Inácio Coelho Sá, Hiran Sátiro Souza da Gama, Ana Paula Silva Oliveira, Rafaela Nunes Dávila, Hélio Afonso Amazonas Júnior, Esly Camico Mandu, Tatiana Bacry Cardoza, Jaqueline de Almeida Gonçalves Sachett, Fernando Almeida-Val, Vinícius Azevedo Machado, Wuelton Marcelo Monteiro, Felipe Leão Gomes Murta.

**Formal analysis:** Alícia Patrine Cacau dos Santos, Theo de Lima Guerra, Evellyn Antonieta Rondon Tomé da Silva, Inácio Coelho Sá, Hiran Sátiro Souza da Gama, Ana Paula Silva Oliveira, Rafaela Nunes Dávila, Hélio Afonso Amazonas Júnior, Tatiana Bacry Cardoza, Fernando Almeida-Val, Vinícius Azevedo Machado, Wuelton Marcelo Monteiro, Felipe Leão Gomes Murta.

**Funding acquisition:** Alícia Patrine Cacau dos Santos, Hiran Sátiro Souza da Gama, Ana Paula Silva Oliveira, Rafaela Nunes Dávila, Hélio Afonso Amazonas Júnior, Fernando Almeida-Val, Wuelton Marcelo Monteiro, Felipe Leão Gomes Murta.

**Investigation:** Alícia Patrine Cacau dos Santos, Theo de Lima Guerra, Evellyn Antonieta Rondon Tomé da Silva, Inácio Coelho Sá, Hiran Sátiro Souza da Gama, Ana Paula Silva Oliveira, Rafaela Nunes Dávila, Hélio Afonso Amazonas Júnior, Joaquim da Silva Lopes, Esly Camico Mandu, Fernando Almeida-Val, Altair Seabra de Farias, Wuelton Marcelo Monteiro, Felipe Leão Gomes Murta.

**Methodology:** Alícia Patrine Cacau dos Santos, Theo de Lima Guerra, Evellyn Antonieta Rondon Tomé da Silva, Inácio Coelho Sá, Hiran Sátiro Souza da Gama, Ana Paula Silva Oliveira, Rafaela Nunes Dávila, Hélio Afonso Amazonas Júnior, Joaquim da Silva Lopes, Esly Camico Mandu, Fernando Almeida-Val, Altair Seabra de Farias, Vinícius Azevedo Machado, Wuelton Marcelo Monteiro, Felipe Leão Gomes Murta.

**Project administration:** Alícia Patrine Cacau dos Santos.

**Supervision:** Theo de Lima Guerra, Evellyn Antonieta Rondon Tomé da Silva, Inácio Coelho Sá, Hiran Sátiro Souza da Gama, Vinícius Azevedo Machado, Wuelton Marcelo Monteiro, Felipe Leão Gomes Murta.

**Validation:** Alícia Patrine Cacau dos Santos, Theo de Lima Guerra, Evellyn Antonieta Rondon Tomé da Silva, Inácio Coelho Sá, Hiran Sátiro Souza da Gama, Ana Paula Silva Oliveira, Rafaela Nunes Dávila, Hélio Afonso Amazonas Júnior, Joaquim da Silva Lopes, Esly Camico Mandu, Jaqueline de Almeida Gonçalves Sachett, Fernando Almeida-Val, Altair Seabra de Farias, Vinícius Azevedo Machado, Wuelton Marcelo Monteiro, Felipe Leão Gomes Murta.

**Visualization:** Theo de Lima Guerra, Evellyn Antonieta Rondon Tomé da Silva, Inácio Coelho Sá, Hiran Sátiro Souza da Gama, Ana Paula Silva Oliveira, Rafaela Nunes Dávila, Hélio Afonso Amazonas Júnior, Joaquim da Silva Lopes, Esly Camico Mandu, Jaqueline de Almeida Gonçalves Sachett, Fernando Almeida-Val, Altair Seabra de Farias, Vinícius Azevedo Machado, Wuelton Marcelo Monteiro, Felipe Leão Gomes Murta.

**Writing – original draft:** Alícia Patrine Cacau dos Santos, Theo de Lima Guerra, Evellyn Antonieta Rondon Tomé da Silva, Inácio Coelho Sá, Hiran Sátiro Souza da Gama, Ana Paula Silva Oliveira, Rafaela Nunes Dávila, Hélio Afonso Amazonas Júnior, Joaquim da Silva Lopes, Esly Camico Mandu, Tatiana Bacry Cardoza, Jaqueline de Almeida Gonçalves Sachett, Fernando Almeida-Val, Altair Seabra de Farias, Vinícius Azevedo Machado, Wuelton Marcelo Monteiro, Felipe Leão Gomes Murta.

**Writing – review & editing:** Alícia Patrine Cacau dos Santos, Theo de Lima Guerra, Evellyn Antonieta Rondon Tomé da Silva, Inácio Coelho Sá, Hiran Sátiro Souza da Gama, Ana Paula Silva Oliveira, Rafaela Nunes Dávila, Hélio Afonso Amazonas Júnior, Joaquim da Silva Lopes, Esly Camico Mandu, Tatiana Bacry Cardoza, Jaqueline de Almeida Gonçalves Sachett, Fernando Almeida-Val, Altair Seabra de Farias, Vinícius Azevedo Machado, Wuelton Marcelo Monteiro, Felipe Leão Gomes Murta.

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
