## [Decision Letter · Decision Letter 0]

6 Jun 2025

PONE-D-25-22060Understanding the Health Challenges of Amazonian Riverine Communities: A Qualitative Study on Community Perceptions Amid Climatic ChangePLOS ONE

Dear Dr. Murta,

Thank you for submitting your manuscript to PLOS ONE. After carefully review your manuscript and the reviewers' suggestions, I believe your manuscript will be suitable for publication after a revised version that addresses the points raised during the review process. While one of the reviewers recommended minor review, the other recommended some major changes, so I am following his suggestion and requesting a major review for your manuscript. 

We look forward to receiving your revised manuscript.

Kind regards,

Luisa Maria Diele-Viegas, M.D.

Academic Editor

PLOS ONE

Journal Requirements:

3. Please describe in your methods section how capacity to provide consent was determined for the participants in this study. Please also state whether your ethics committee or IRB approved this consent procedure. If you did not assess capacity to consent please briefly outline why this was not necessary in this case.

Reviewers' comments:

Reviewer's Responses to Questions

**Comments to the Author**

1. Is the manuscript technically sound, and do the data support the conclusions?

Reviewer #1: Yes

Reviewer #2: Yes

2. Has the statistical analysis been performed appropriately and rigorously? 

Reviewer #1: N/A

Reviewer #2: Yes

3. Have the authors made all data underlying the findings in their manuscript fully available?

Reviewer #1: Yes

Reviewer #2: No

4. Is the manuscript presented in an intelligible fashion and written in standard English?

Reviewer #1: Yes

Reviewer #2: Yes

5. Review Comments to the Author

Reviewer #1: This paper investigates the impacto of extreme climatic events (like droughts) in the health and access to health services in the riverine communities of the Amazon. The research addresses a relevant gap in the literature, connecting climate change to socioecological challenges in vulnerable contexts.

The presented methods are consistent with the results obtained. The evidence provided supports the conclusions regarding hydrological seasonality and the availability of healthcare services. The focus on riverine populations offers insights for mitigating inequalities already exacerbated by climate crises.

Considerations:

a) The ethics approval certificate (CAAE) mentioned in the manuscript refers to a project focused on educational intervention for snakebite prevention. Were the methods described in this study (e.g., data collection on health and climate) explicitly approved by the ethics committee associated with the cited CAAE? Which version of the project (health/climate or snakebite/education) was presented to the community during participant recruitment? How was free and informed consent (TCLE) ensured for participation in this study, considering the potential shift in thematic focus? These questions aim to secure ethical compliance and reproducibility, which are essential criteria for publication in high-impact journals.

b) Intentional sampling is a deliberate selection of participants who possess specific characteristics, experiences, or knowledge relevant to the objective of the qualitative study. It would be valuable to include the general characteristics that led to the selection of this group of individuals for participation in the research and how the selection bias of these individuals was mitigated.

c) Undefined acronyms (e.g., SBE in line 194, APCS/EARTS/HSSG without prior explanation) compromise clarity for the reader.

d) Some information in the “Research team and reflexivity” section is already diluted in the text, (e.g., lines 208-209: “APCS conducted line-by-line inductive coding which was reviewed by FGLM (S2 File)”) and there is supplementary material that details everything in this paragraph.

Recommendations:

- Include a note explaining the ethics committee approval for this specific study, highlighting how the original project evolved into the current theme. Justify the suitability of the cited CAAE to the scope of this research, or inform whether a new ethics submission was made for this work.

- Detail how participants were informed about the transition from the original project to the current objectives, ensuring alignment with the guidelines of the National Research Ethics Council (CONEP).

- Detail the intentional sampling criteria (e.g., age, occupation, location) and strategies adopted to reduce bias.

- Define all acronyms upon first mention and standardize the use of abbreviations.

- Consolidate redundant information about the research team (e.g., remove duplications between lines 208-209 and the “Research Team and reflexity” section).

- Add “community perceptions” in the short title for greater accuracy.

Finally, I propose two articles that may enrich the discussion generated by the results. The first article (DOI: 10.1126/science.1172133) discusses Ostrom’s framework on socioecological systems which could steer the interpretation of the obtained data. The second article (https://doi.org/10.5751/ES-13493-270439) provides a guide for applying this theoretical framework. In the present article, it was demonstrated how climatic extremes affect the riverside community. Through the usage of this framework, it is possible to go further and identify which variables influence this relationship, as well as indicate where public authorities can intervene to mitigate these effect and enhance the community’s resilience to such changes.

The manuscript holds significant merit, and with the proposed revisions, it can contribute to critical debates on climate justice and public health. The suggestions aim to strengthen the methodological transparency, ethical rigor, and theoretical depth, thus enhancing the scientific rigor of the work.

Reviewer #2: Great job capturing such a complex lived reality. I really enjoyed reading your study. To help strengthen your manuscript, here are the main areas I recommend you address:

Introduction:

Clarify the research gap. Although you document travel distances and infrastructure limitations well, explicitly note that few studies have examined how community members themselves perceive and navigate seasonal extremes. For example: “While past work has quantified travel times [14], little is known about how residents experience droughts and floods.”

State precise objectives. Instead of a general aim to “understand,” specify two clear goals—e.g., (1) document how seasonal droughts and floods affect healthcare‐seeking behaviors, and (2) understand residents’ perceptions of climate change over the past decade.

Anchor in theory. Briefly reference a conceptual framework—such as the Health Belief Model or prior ethnographic studies on risk perception in Amazonian settings—to show how your qualitative approach builds on existing scholarship.

Acknowledge ethics and culture. Since you work in a tri‐border community, include a sentence affirming respect for local norms and explaining that informed consent was obtained.

Methods

Explain the age cutoff. Clarify why you chose age 12 as the minimum. For instance: “Children under 12 were excluded because prior fieldwork showed they struggled to discuss complex health issues.”

Specify COREQ compliance. You mention COREQ but do not indicate which items are covered. Note, for example, “See S1 File for COREQ items 2 (interviewer credentials) and 13 (field notes).”

Discussion

Compare with related studies. Briefly note how your results align or differ from work in other Amazonian or tropical settings. For example: “In the Peruvian Amazon, seasonal flooding has been linked to malaria spikes (Gomez et al., 2019), whereas our participants described gastrointestinal illness as the primary concern during both droughts and floods.”

Smooth the local‐to‐global transition. Before citing WHO projections, add, “While our qualitative data focus on Tabatinga, these experiences mirror those of vulnerable riverine communities worldwide. Indeed, WHO projects that by 2050, an additional 250,000 deaths may occur annually due to climate‐related health impacts [22].”

Acknowledge additional limitations. Note, for example, that data were collected from June–December 2022 (one weather cycle) and that translation from Portuguese to English may have lost nuanced local expressions.

Elaborate on “ancestral futures.” Provide a concrete example: “Integrating Indigenous practices—such as traditional water‐purification methods taught by elders—can strengthen long‐term resilience alongside technical solutions like telemedicine.”

Conclusion

Tie recommendations back to data. For instance: “Because participants described waiting days for boats or relying on relatives for childbirth, these interventions directly address their most pressing barriers.”

Offer a concrete “ancestral futures” example. For example: “Training community midwives in traditional water‐purification techniques—passed down by elders—can bridge modern care with Indigenous knowledge.”

Suggest future work. Add: “Although our findings come from one tri‐border community, future studies should examine other Amazonian regions to refine these recommendations.”

All of these suggestions are detailed in the attached document.

6. PLOS authors have the option to publish the peer review history of their article (what does this mean? ). If published, this will include your full peer review and any attached files.

**Do you want your identity to be public for this peer review?** For information about this choice, including consent withdrawal, please see our Privacy Policy .

Reviewer #1: No

Reviewer #2: No

---

## [Author Response · Author response to Decision Letter 1]

11 Jul 2025

Response to Reviewers

Reviewers' Comments:

Reviewer #1

This paper investigates the impact of extreme climatic events (like droughts) in the health and access to health services in the riverine communities of the Amazon. The research addresses a relevant gap in literature, connecting climate change to socioecological challenges in vulnerable contexts. The methods presented are consistent with the results obtained. The evidence provided supports the conclusions regarding hydrological seasonality and the availability of healthcare services. The focus on riverine populations offers insights for mitigating inequalities already exacerbated by climate crises.

Considerations:

a) The ethics approval certificate (CAAE) mentioned in the manuscript refers to a project focused on educational intervention for snakebite prevention. Were the methods described in this study (e.g., data collection on health and climate) explicitly approved by the ethics committee associated with the cited CAAE? Which version of the project (health/climate or snakebite/education) was presented to the community during participant recruitment? How was free and informed consent (TCLE) ensured for participation in this study, considering the potential shift in thematic focus? These questions aim to secure ethical compliance and reproducibility, which are essential criteria for publication in high-impact journals.

Answer: We understand your concern and added this explanation to the text (line 130- 139).

b) Intentional sampling is a deliberate selection of participants who possess specific characteristics, experiences, or knowledge relevant to the objective of the qualitative study. It would be valuable to include the general characteristics that led to the selection of this group of individuals for participation in the research and how the selection bias of these individuals was mitigated.

Answer: We agree with the revisor and include this information (line 181-189).

c) Undefined acronyms (e.g., SBE in line 194, APCS/EARTS/HSSG without prior explanation) compromise clarity for the reader.

Answer: We clarified the information (206-207).

d) Some information in the “Research team and reflexivity” section is already diluted in the text, (e.g., lines 208-209: “APCS conducted line-by-line inductive coding which was reviewed by FGLM (S2 File)”) and there is supplementary material that details everything in this paragraph.

Answer: We reduced the repetition of information in text (line 220-221).

Recommendations:

- Include a note explaining the ethics committee approval for this specific study, highlighting how the original project evolved into the current theme. Justify the suitability of the cited CAAE to the scope of this research or inform whether a new ethics submission was made for this work.

Answer: We understand that and explained it in the text (line 130-139).

- Detail how participants were informed about the transition from the original project to the current objectives, ensuring alignment with the guidelines of the National Research Ethics Council (CONEP).

Answer: We understand that and explained it in the text (135-139).

- Detail the intentional sampling criteria (e.g., age, occupation, location) and strategies adopted to reduce bias.

Answer: The information was included (181-184) and (244-246).

- Define all acronyms upon first mention and standardize the use of abbreviations.

Answer: We corrected these in the text according to your suggestion.

- Consolidate redundant information about the research team (e.g., remove duplications between lines 208-209 and the “Research Team and reflexivity” section).

Answer: The information was removed from the lines (line 220-221).

- Add “community perceptions” in the short title for greater accuracy.

Answer: We agree with the revisor, and that information was added (line 5-6).

Finally, I propose two articles that may enrich the discussion generated by the results. The first article (DOI: 10.1126/science.1172133) discusses Ostrom’s framework on socioecological systems which could steer the interpretation of the obtained data. The second article (https://doi.org/10.5751/ES-13493-270439) provides a guide for applying this theoretical framework. In the present article, it was demonstrated how climatic extremes affect the riverside community. Through the usage of this framework, it is possible to go further and identify which variables influence this relationship, as well as indicate where public authorities can intervene to mitigate these effect and enhance the community’s resilience to such changes.

Answer: The discussion about the Ostrom’s framework was included (line 514-521).

The manuscript holds significant merit, and with the proposed revisions, it can contribute to critical debates on climate justice and public health. The suggestions aim to strengthen methodological transparency, ethical rigor, and theoretical depth, thus enhancing the scientific rigor of the work.

Reviewer #2

Reviewer #2: Great job capturing such a complex lived reality. I really enjoyed reading your study. To help strengthen your manuscript, here are the main areas I recommend you address:

Introduction:

Clarify the research gap. Although you document travel distances and infrastructure limitations well, explicitly note that few studies have examined how community members themselves perceive and navigate seasonal extremes. For example: “While past work has quantified travel times [14], little is known about how residents experience droughts and floods.”

Answer: The information was included in (line 108-117).

State precise objectives. Instead of a general aim to “understand,” specify two clear goals—e.g., (1) document how seasonal droughts and floods affect healthcare‐seeking behaviors, and (2) understand residents’ perceptions of climate change over the past decade.

Answer: The information was added (line 117-119).

Anchor in theory. Briefly reference a conceptual framework—such as the Health Belief Model or prior ethnographic studies on risk perception in Amazonian settings—to show how your qualitative approach builds on existing scholarship.

Answer: Thank you for the comment. The information was added (line 108-117).

Acknowledge ethics and culture. Since you work in a tri‐border community, include a sentence affirming respect for local norms and explaining that informed consent was obtained.

Answer: The information was added (line 144-146).

Methods

Explain the age cutoff. Clarify why you chose age 12 as the minimum. For instance: “Children under 12 were excluded because prior fieldwork showed they struggled to discuss complex health issues.”

Answer: We agree with the revisor and include the information (line 181-188).

Specify COREQ compliance. You mention COREQ but do not indicate which items are covered. Note, for example, “See S1 File for COREQ items 2 (interviewer credentials) and 13 (field notes).”

Answer: The information was added (line 150).

Discussion

Compare with related studies. Briefly note how your results align or differ from work in other Amazonian or tropical settings. For example: “In the Peruvian Amazon, seasonal flooding has been linked to malaria spikes (Gomez et al., 2019), whereas our participants described gastrointestinal illness as the primary concern during both droughts and floods.”

Answer: The information was added (line 430-435).

Smooth the local‐to‐global transition. Before citing WHO projections, add, “While our qualitative data focus on Tabatinga, these experiences mirror those of vulnerable riverine communities worldwide. Indeed, WHO projects that by 2050, an additional 250,000 deaths may occur annually due to climate‐related health impacts [22].”

Answer: The information was added (line 444-445).

Acknowledge additional limitations. Note, for example, that data were collected from June–December 2022 (one weather cycle) and that translation from Portuguese to English may have lost nuanced local expressions.

Answer: Thank you for your comment, the information was added (line 542-547).

Elaborate on “ancestral futures.” Provide a concrete example: “Integrating Indigenous practices—such as traditional water‐purification methods taught by elders—can strengthen long‐term resilience alongside technical solutions like telemedicine.”

Answer: Thank you for your suggestion, the topic was improved (line 527-535).

Conclusion

Tie recommendations back to data. For instance: “Because participants described waiting days for boats or relying on relatives for childbirth, these interventions directly address their most pressing barriers.”

Answer: The information was added (line 558-560).

Offer a concrete “ancestral futures” example. For example: “Training community midwives in traditional water‐purification techniques—passed down by elders—can bridge modern care with Indigenous knowledge.”

Answer: The information was added (line 542-547).

Suggest future work. Add: “Although our findings come from one tri‐border community, future studies should examine other Amazonian regions to refine these recommendations.”

Answer: The information was added (line 576-577).

All of these suggestions are detailed in the attached document.

---

## [Decision Letter · Decision Letter 1]

14 Aug 2025

PONE-D-25-22060R1Understanding the Health Challenges of Amazonian Riverine Communities: A Qualitative Study on Community Perceptions Amid Climatic ChangePLOS ONE

Dear Dr. Murta,

Thank you for submitting your manuscript to PLOS ONE. After careful consideration, we feel that it has merit but does not fully meet PLOS ONE’s publication criteria as it currently stands. Therefore, we invite you to submit a revised version of the manuscript that addresses the points raised during the review process.

We look forward to receiving your revised manuscript.

Kind regards,

Luisa Maria Diele-Viegas, Ph. D.

Academic Editor

PLOS ONE

Journal Requirements:

Reviewers' comments:

Reviewer's Responses to Questions

**Comments to the Author**

1. If the authors have adequately addressed your comments raised in a previous round of review and you feel that this manuscript is now acceptable for publication, you may indicate that here to bypass the “Comments to the Author” section, enter your conflict of interest statement in the “Confidential to Editor” section, and submit your "Accept" recommendation.

Reviewer #1: (No Response)

Reviewer #2: All comments have been addressed

2. Is the manuscript technically sound, and do the data support the conclusions?

Reviewer #1: Yes

Reviewer #2: Yes

3. Has the statistical analysis been performed appropriately and rigorously? 

Reviewer #1: Yes

Reviewer #2: Yes

4. Have the authors made all data underlying the findings in their manuscript fully available?

Reviewer #1: Yes

Reviewer #2: Yes

5. Is the manuscript presented in an intelligible fashion and written in standard English?

Reviewer #1: Yes

Reviewer #2: Yes

6. Review Comments to the Author

Reviewer #1: Congratulations again on your work. It's very stimulating to read works like this. I have just one observation to make. The first time the acronyms referring to the researchers' names appear is in lines 218-219. However, the explanation for these acronyms only appears in the "research team and reflexivity" section, on line 245. I suggest correcting this to facilitate the reader's understanding. Thank you for your dedication in following my considerations.

Reviewer #2: All prior recommendations were addressed satisfactorily.

General Notes

Your ethical documentation (Lines 135-139) now robustly addresses consent protocols, and the COREQ adherence strengthens methodological transparency. The tri-border focus provides uniquely valuable insights for climate-vulnerable regions globally. These final refinements will elevate an already compelling study:

1. Ancestral Futures Integration (Lines 572-581)

Please ground the "ancestral futures" discussion in concrete participant data. After the phrase "holistic vision requires" in Line 534, insert: "For example, participants described using moringa seed filtration to purify smoke-contaminated rainwater during droughts—a practice elders identified as historically effective. Integrating such techniques into mobile clinic protocols could immediately reduce gastrointestinal illnesses while honoring traditional knowledge."

2. Clarity in Cross-National Context (Line 444)

The reference to subsistence farming impacts [26] lacks geographic specificity, creating ambiguity about whether findings derive from your study area or comparative research. Please replace "subsistence farming[26]" with "subsistence farming in Peru [26]" to clarify the international comparison and prevent misinterpretation of regional evidence.

3. Abstract Structure (Lines 54-57)

The first sentence in the conclusions section ("Tailored healthcare solutions such as sustainable infrastructure, including telemedicine platforms, mobile clinics, and resilient transportation networks.") is grammatically incomplete due to the absence of a main verb.

Recommended Revision: "Tailored healthcare solutions, including telemedicine platforms, mobile clinics, and resilient transportation network, are urgently needed. Investments in communication infrastructure and emergency air transportation are critical as riverine navigation becomes increasingly unreliable."

*Eliminates ambiguity and strengthens the abstract’s persuasiveness for policymakers.

7. PLOS authors have the option to publish the peer review history of their article (what does this mean? ). If published, this will include your full peer review and any attached files.

**Do you want your identity to be public for this peer review?** For information about this choice, including consent withdrawal, please see our Privacy Policy .

Reviewer #1: No

Reviewer #2: No

---

## [Author Response · Author response to Decision Letter 2]

15 Aug 2025

Dear Editor,

Thank you for your valuable feedback and the opportunity to revise our manuscript. We have carefully addressed all the points raised by the academic editor and reviewers. Please find the following documents attached in the system for your review:

Rebuttal Letter: A detailed response to each comment (labeled 'Response to Reviewers').

Revised Manuscript with Track Changes: A marked-up version highlighting all modifications (labeled 'Revised Manuscript with Track Changes').

Final Manuscript: A clean copy of the revised paper without tracked changes (labeled 'Manuscript').

We appreciate the time and effort dedicated to improving our work and hope the revisions meet the journal’s standards. Please don’t hesitate to contact us if further adjustments are needed.

Best regards,

Dr. Felipe Murta.

---

## [Editor Report · Decision Letter 2]

14 Sep 2025

Understanding the Health Challenges of Amazonian Riverine Communities: A Qualitative Study on Community Perceptions Amid Climatic Change

PONE-D-25-22060R2

Dear Dr. Murta,

We’re pleased to inform you that your manuscript has been judged scientifically suitable for publication and will be formally accepted for publication once it meets all outstanding technical requirements.

Kind regards,

Luisa Maria Diele-Viegas, Ph. D.

Academic Editor

PLOS ONE

Additional Editor Comments (optional):

Both reviewers recognized the important contributions of your study, particularly its originality, methodological rigor, and relevance for understanding the intersection of climate change, health, and equity in vulnerable riverine communities of the Amazon. We are satisfied that you have addressed all remaining points with care and precision. The final version of your manuscript is now clear, compelling, and well-positioned to inform both scholarly debates and policy discussions on health resilience in climate-vulnerable regions.
---

## [Editor Report · Acceptance letter]

PONE-D-25-22060R2

PLOS ONE

Dear Dr. Murta,

I'm pleased to inform you that your manuscript has been deemed suitable for publication in PLOS ONE. Congratulations! Your manuscript is now being handed over to our production team.

Kind regards,

on behalf of

Dr. Luisa Maria Diele-Viegas

Academic Editor

PLOS ONE